# Photothermal and Catalytic Performance of Multifunctional Cu-Fe Bimetallic Prussian Blue Nanocubes with the Assistance of Near-Infrared Radiation

**DOI:** 10.3390/nano13131897

**Published:** 2023-06-21

**Authors:** Bairui Qi, Qiang Xu, Yunxuan Cao, Zhu Xiao

**Affiliations:** 1School of Materials Science and Engineering, Central South University, Changsha 410083, China; 2Key Laboratory of Non-Ferrous Metal Materials Science and Engineering, Ministry of Education, Changsha 410083, China

**Keywords:** Prussian blue, photothermal effects, catalytic reaction, synergistic action, copper

## Abstract

Copper and iron are the basic metal elements that have attracted much attention in industry. Prussian blue (PB) is a significant class of metal–organic frameworks (MOFs); however, the lack of such linkages between the structure and properties, as well as properties differences, limits their potential applications. In this paper, the Cu-based Prussian blue nanocubes with and without Fe doping were synthesized. With the increasing reaction time, the morphology of the Cu-based Prussian blue nanocubes without Fe doping (PB:Cu NCs) changes from cuboidal to circular, and finally grows back to cuboidal. However, Cu-based Prussian blue nanocubes with Fe doping (PB: CuFe NCs) grow directly from the cube and eventually collapse. The nanocubes show a notable red shift with the tunable spectra from 400 nm to 700 nm. Compared with PB: Cu NCs, the PB: CuFe NCs have higher temperature rise under 808 nm irradiation and better photothermal efficacy. The catalytic efficiency of PB: CuFe NCs changes with the pH and reaches its maximum value of 1.021 mM with a pH of 5.5. The enhanced catalytic reaction by the near-infrared radiation plasmonic photothermal effect is also confirmed. This work highlights the potential of the developed PB: Cu and PB: CuFe NCs for photothermal-enhanced co-catalysis nanomaterials.

## 1. Introduction

Copper and iron are vital elements that are important for several critical physiological processes, and have been widely used in the catalytic and biological industry [1,2]. Due to their various characteristics including the inherent high strength [3], high conductivity [4], large energy storage capacity [4], biosafety [5], and nontoxicity [6], copper and iron can boost performance by deliberate design, acting as promising nanomaterials. However, the relationship between these properties of copper and iron at nanoscale and their corresponding mechanisms are still on their way to be developed.

Metal–organic frameworks (MOFs) are promising due to their structural tunability, highly modular manner, and ordered arrangements of multi-photonic properties [7]. In particular, MOFs with optimal photonic functionality present a bright future [8] because they allow for light-harvesting capacity manipulation [9]. Cyanide-bridged bimetallic systems known as Prussian blue (PB) are a significant class of MOFs, and they are fascinating in many research areas [10]. PB is one of the low-cost medicines for thallotoxicosis, and it has been approved by the United States Food and Drug Administration (FDA) [11]. Numerous scientists have been interested in PB due to its versatility as a research tool and its high biosafety [12,13], compatibility [14,15], and accessible surface functional modification capabilities [5,16]. Additionally, PB demonstrates a wide photothermal absorption range, which is related to a charge-transfer transition between Fe^2+^ and Fe^3+^ [17]. However, its poor photothermal efficiency due to the low photothermal conversion efficiency limits its photothermal application. Therefore, it is essential to optimize the design of PB derivatives with tunable optical characteristics by varying primary atoms and combining diverse capabilities [18,19]. Studies have shown that the free carrier concentrations of nanostructures are the primary factor to determine light absorption and reflection, and the increase in carrier concentration of the carrier often leads to increased and adjustable spectrum absorption [20]. For Fe-based PB, the doping of other ions in PB is often helpful to improve its optical, electrical, and other properties [21]. Cu^2+^ has a different ionic radius and crystallographic characteristics to Fe^2+^. However, there is little work reported on the effect of Cu^2+^ and the combined action of Cu^2+^ and Fe^2+^ on the absorption range and optical properties of PBs. It would be quite meaningful to develop Cu-based PB derivatives with adjustable optical characteristics and improved photothermal performance.

The generation of reactive oxygen species (ROS) is one of the critical mechanisms of catalytic reaction [22]. The ability of PB to scavenge ROS is due to its multienzyme-like activity (e.g., peroxidase (POD), catalase (CAT), and superoxide dismutase (SOD)) [18,23]. PB as a POD mimic can suppress the production of hydroxyl radicals (•OH) during the catalytic reaction [24]. It implies that specific PB nano-objects can be used as nanocatalysts, nanocarriers, or nanodrugs [11,25,26]. Newly disclosed nano-objects of the PB type are multifunctional, combining several capacities [27,28], and they allow the connection of many types of catalysts or cells in a single nanoparticle [29,30]. The use of PB could be enhanced with further advancements in its applications. The photothermal and catalytic capabilities of PB can be combined to generate reciprocal synergistic action among these unique properties, resulting in the enhanced full utilization [31].

In this work, Cu-based Prussian blue nanocubes (PB: Cu NCs) and Fe-doped Prussian blue nanocubes (PB: CuFe NCs) were synthesized using a straightforward hydrothermal technique, supplementing the reaction directly with copper and divalent iron sources, respectively. In the PB: Cu NCs’ crystal structure, Fe^2+^ substitutes Cu^2+^ completely, resulting in a red-shifted absorption spectrum that allows for the control of its optical characteristics. Photothermal efficiency also establishes the connection of optical absorption between PB: Cu and PB: CuFe NCs. The chromogenic agent OPD was used to assay the POD-like enzyme activity of PB: CuFe NCs, and the synergetic photothermal/catalysis performance was further verified. In the nano-era, the PB system in this study offers a viable alternative approach for nano-photothermal co-catalysis materials.

## 2. Materials and Methods

Potassium ferricyanide (K_3_[Fe(CN)_6_]), iron(II) chloride (FeCl_2_), copper(II) chloride (CuCl_2_), and dimethyl sulfoxide were supplied by Aladdin Chemistry Co. Ltd. (Shanghai, China). Poly(vinylpyrrolidone) (PVP K30) and hydrochloric acid (HCl) were obtained from Sinopharm Chemical Reagent Co., Ltd. (Beijing, China). O-Phenylenediamine (OPDA) was received from Macklin Biochemical Technology Co., Ltd. (Shanghai, China). Copper and iron standard solutions were obtained from Guobiao Testing and Certification Co., Ltd. (Beijing, China). All chemicals and solvents were of analytical grade and used as received. 

### 2.1. Synthesis of PB: Cu and PB: CuFe NCs

A 20 mL mixture solution containing FeCl_2_, CuCl_2_, PVP, and HCl was prepared, with a HCl concentration of 2 mmol. The solution was then heated to the temperature of 80 °C. After stirring the solution for 5 min, a 20 mL aqueous solution containing 0.1 mmol of K_3_[Fe(CN)_6_, 1 g of PVP, and 2 mmol of HCl was added. The finished solution was stirred for 1 min at 80 °C and then cooled to room temperature. When the added molar ratio of FeCl_2_ and CuCl_2_ was 0:0.15, PB: Cu NCs were obtained. When the added molar ratio FeCl_2_ and CuCl_2_ was added to 0.11:0.4, PB: CuFe NCs were obtained. The precipitate was produced by centrifugation in acetone solution, and then washed by acetone and deionized water twice.

### 2.2. Characterization

The emission scanning electron microscope (SEM) images were taken using a 20 kV MIRA4 LMH field emission SEM (TESCAN, Brno, Czech). Transmission electron microscopy (TEM) images were observed with a FEI Tecnai F30 transmission electron microscope (FEI, Hillsboro, OR, USA) operating at 100 kV. Crystal structures were determined with the aid of a SmartLab3 kW X-ray diffractometer (XRD) with Cu Kα radiation (λ = 1.5406 Å) at 40 kV and 30 mA (Rigaku, Tokyo, Japan). The Renishaw inVia Raman microscope (Renishaw, UK) and UV-3600iPLUS UV-vis-NIR spectrophotometer (Shimadzu Scientific Instruments, Kyoto, Japan) were used to acquire the Raman spectra and the extinction spectra, respectively. Nicolet iS50 (Thermo Fisher Scientific, Waltham, MA, USA) FTIR spectra were obtained for this study. XPS test was performed on Thermo Scientific K-Alpha using an Al Kα excitation source at 6 mA and 12 kV (Thermo Fisher Scientific, USA). A TCP-5100-VDV inductively coupled plasma–mass spectrometer (ICP-MS) was used to analyze the composition quantitatively (NYSE: A, Malaysia). Malvern Zetasizer Nano ZSE analyzer was used to determine the hydrodynamic size and zeta potential (Malvern, UK).

### 2.3. Photothermal Performance of PB: Cu and PB: CuFe NCs

Aqueous dispersions of PB: Cu and PB: CuFe NCs with different concentrations, 25, 50, 100, and 200 μg mL^−1^, were examined for their photothermal performance, respectively. PB: Cu and PB: CuFe NCs were heated with a continuous 808 nm laser (1 W cm^–2^) for 10 min, and the resulting temperatures were captured with an infrared thermal imaging camera (GUIDE INFRARED, Wuhan, China). The photothermal stability of PB: Cu and PB: CuFe NCs was measured after five laser on/off cycles. After cooling to room temperature from an initial temperature of 37 °C, a suspension of PB: Cu and PB: CuFe NCs was subjected to 10 min illumination at 808 nm (1 W cm^–2^). The photothermal conversion efficiency (*η*) of PB: Cu and PB: CuFe NCs was determined using [32]:(1)$$\eta =\frac{hS(T_{Max}-T_{surr})-Q_{Dis}}{I(1-10^{-A808})}$$
where *T_max_* and *T_surr_* are the maximum and standard temperatures, respectively. The absorbance of the solutions at 808 nm (shown by A808) is proportional to the laser power density represented by *I*. *Q_Dis_*, and the value of 5.4 × 10^−4^ × *I* (J/s) is the dissipated energy of the test solution due to the solvent absorption when using an 808 nm laser. The time (τ) taken for the laser to cool down can be obtained using:(2)$$\tau =\frac{t}{-ln(\frac{T-T_{Surr}}{T_{Max}-T_{surr}})}$$
(3)$$hS=\frac{{mC}_{p}}{\tau }$$
where *S* is the surface area of the laser spot, *h* is the heat transfer coefficient, *m* is the mass of the PB: Cu and PB: CuFe NCs, and *C_p_* is its heat capacity (4.2 J/(g·°C)), respectively. 

### 2.4. Steady-State Kinetic Assays

The absorbance of the PB: CuFe NCs was recorded in time-course mode at 25 °C using a UV-vis-NIR spectrophotometer for the steady-state kinetic experiments. At the same time, kinetic tests of PB: CuFe NCs were conducted under the condition of different concentrations of H_2_O_2_ as the substrate (0.25, 0.5, 0.5, 1, and 2 mM). After nonlinear least-squares fitting of the absorbance data using the Michaelis–Menten equation, the apparent kinetic parameters were obtained using [33]: (4)$$V_{0}=\frac{V_{max}\cdot [S]}{K_{m}+[S]}$$

The maximum velocity (*V_max_*) and Michaelis–Menten constant (*K_m_*) were calculated according to the linear double-reciprocal plot (Lineweaver–Burk plot):(5)$$\frac{1}{V_{0}}=\frac{K_{m}}{V_{max}}\cdot \frac{1}{\left[S\right]}+\frac{1}{V_{max}}$$
where *v* stands for the initial reaction rate, *V_max_* represents the maximum conversion rate, [*S*] represents the concentration of the substrate, and *K_m_* is the Michaelis constant, respectively.

## 3. Results and Discussion

### 3.1. Preparation and Characterization of PB:Cu and PB:Fe NCs

The PB: Cu and PB: CuFe NCs were prepared using simple hydrothermal method with polyvinylpyrrolidone (PVP) as the stabilizer (Figure 1A). As a typical iron ferrocyanide, PB has a cubic cell size of 10.2 Å and is composed of iron (III) ions coupled with nitrogen atoms and iron (II) ions coordinated with carbon atoms. Low-spin and high-spin electron configurations were formed in each PB network unit by the iron (II) bonded to the cyanogen carbon atom and the iron (III) surrounded by the cyanogen nitrogen terminal. Figure 1B,G reveal that the PB: Cu and PB: CuFe NCs after 1 min reaction show a cubic shape, with the average size of 100 nm and 50 nm, respectively. Transmission electron microscopy (TEM) of PB: CuFe NCs revealed a solid cubic form with a smooth surface. Selected area electron diffraction (SAED) patterns confirmed the fcc structure of PB: CuFe NCs (Figure 1L). After a 30 min reaction, most of the nanocubes still maintained primarily cube-shape and uniform sizes; the edge of some PB: Cu NCs became much smoother, suggesting that this stage is the growing stage of PB: CuFe NCs (Figure 1C,H). Prolonging the reaction period to 60 min, the size of the nanocubes continued to increase, and the PB: Cu NCs were much more spherical (Figure 1D) than the PB: CuFe NCs (Figure 1D,I). After reacting for 120 min, almost all of the PB: Cu NCs had the rounded cube shape, and their morphology became uneven (Figure 1E). Most of the PB: CuFe NCs were damaged, showing an irregular shape (Figure 1J). Increasing the reaction time to 240 min, some PB: Cu nanocubes with large particle sizes appeared, showing a more obvious cavity structure on the surfaces of the cubes (Figure 1F). The PB: CuFe NCs were wholly broken (Figure 1K). This approach may create smaller nanocubes in a relatively brief period, and the growth process of PB: Cu and PB: CuFe NCs for 240 min can be observed. Throughout the reaction, the PB: Cu NCs gradually evolved from a nanocube to a rounded cube with a large size, and the surface was also etched to reveal a void-like structure. The particle size of PB: CuFe NCs gradually increased while the cubic shape was maintained, and it eventually collapsed and cracked with the increasing reaction time. Therefore, the relevant research and analysis of PB: Cu and PB: CuFe NCs with a reaction time of 1min were focused on for further investigation.

The detailed crystallite parameters of PB: Cu and PB: CuFe NCs were acquired through X-ray diffraction (XRD) measurement. The diffraction peaks at 2*θ* = 17.3°, 24.6°, 35.2°, and 39.6° correspond to (100), (110), (200), and (210) facets (Figure 2A), respectively [15]. This result demonstrates that the peak intensity of PB: Cu NCs is significantly greater than that of PB: CuFe NCs, indicating that PB: Cu particles possess good crystallinity and relatively large grain size, which is consistent with the SEM results in Figure 1. The optical properties of PB: Cu and PB: CuFe NCs were investigated using the UV-vis-NIR spectrum. PB: CuFe NCs exhibited a prominent absorption peak at approximately 700 nm (Figure 2B) due to the charge transfer between Fe^2+^ and Fe^3+^ ions. However, PB: Cu NCs exhibited only a weak absorption peak at approximately 400 nm (Figure 2B), suggesting that the optical properties of PB undergo a significant redshift, and the prominent absorption peak appears entirely when Cu^2+^ ions are replaced by Fe^2+^ ions. Studies have shown that it is possible to convert PB: Cu NCs to PB: CuFe NCs by substituting Cu^2+^ ions with Fe^2+^ ions. The net electron band gap reduction effect results in a reduced band gap of the PB: Cu NCs. The optical properties are usually influenced by the band structure changes in PB: Cu and PB: CuFe NCs. Notably, the bands of the host lattice and its interaction with the crystal structure determine the PB: Cu and PB: CuFe NCs’ properties [34]. Hence, substituting Cu^2+^ ions to modify the carrier concentration by replacing Fe^2+^ has a dramatic influence on the corresponding optical characteristics. Simultaneously, Raman and FTIR spectra indicate that the peaks at 3630, 2082, 1606, and 602 nm correspond to O-H, C≡N, Fe^3+^-C≡N-M^2+^, O-H and M^2+^-CN bonds, respectively [12,35] (Figure 2C,E). Normally, with the increase in the laser exposure time, the Raman signal strength of PB: CuFe NCs keeps constant, but that of PB: Cu NCs changes slightly. The subsequent changes in FTIR and Raman spectra are consistent with each other, indicating that both PB: Cu and PB: CuFe particles have the Fe-CN–metal structure. Moreover, there are more evident PB structures in PB: Cu NCs than those in PB: CuFe NCs, which is consistent with the XRD results. Figure 2D illustrates the surface charges of PB: Cu and PB: CuFe NCs. It is noticed that both particles have the negative zeta potentials; a value of −21.1 for PB: Cu particles and −30.3 mV for PB: CuFe particles, respectively (Figure 2D). This suggests that both PB: Cu and PB: CuFe particles have high colloidal stability.

The hydrodynamic size of nanoparticles in an aqueous coating solution was used to determine the dispersion stability. PB: Cu and PB: CuFe NCs have average sizes of 180.00 ± 5.29 nm and 118.17 ± 2.33 nm, respectively (according to Figure 3A). The results show that the hydrodynamic size of PB: Cu NCs is larger than that of PB: CuFe NCs, which is consistent with the SEM results. The chemical characteristics of the surface of PB: Cu and PB: CuFe NCs were investigated using XPS (Figure 3B). The binding energies of Fe^3+^ obtained from the Fe_4_[Fe(CN)_6_]^3−^ of Fe 2p_3/2_ and Fe 2p_1/2_ were determined to be 711.3 and 707.4 eV, respectively, and the peak at 707.4 eV was defined as Fe 2p_3/2_ at [Fe(CN)_6_]^4−^. The spectra of Cu 2p reveal two doublet peaks (932.2 and 952.4 eV) and satellite peaks (583 and 942.47 eV), which could correspond to Cu 2p_3/2_ and Cu 2p_1/2_, respectively. The results show that Cu ions with the +2 state and Fe ions with the +2 and +3 state in Cu NCs and PB: CuFe NCs. Intriguingly, there are few relevant peaks corresponding to Fe^2+^, and only Fe^3+^ peak remains in PB: Cu NCs Fe 2p spectra. When the copper ions (CuCl_2_) are replaced by the iron ions (FeCl_2_) during the reaction, the intensity of the Fe^2+^ peak increases, while that of the Fe^3+^ peak decreases. According to the above results, the PB: Cu NCs were successfully synthesized using this approach, and it could be safely suspected that Fe^2+^ might replace Cu^2+^ to form the PB: CuFe NCs without the change in their structures.

### 3.2. Photothermal and Properties of PB:Cu and PB:Fe NCs

Nanomaterials with a high NIR absorption are well known to benefit considerably from the laser-induced photothermal effect. Because of their capacity to convert light into heat, PB: Cu and PB: CuFe NCs are ideal candidates as photothermal agents. In this study, PB: Cu and PB: CuFe NCs were subjected to an 808 nm laser at a power density of 0.5, 1.0, 1.5, and 2.0 W cm^−2^. With the increase in power, it was noticed that the highest temperature of about 82.3 °C was achieved by PB: CuFe NCs. However, PB: Cu NCs exhibited just a minor temperature rise, which is much lower than the temperature rise in PB: CuFe NCs (Figure 4A,D). After irradiating the solution with an 808 nm laser (1.0 W cm^2^, 10 min), different concentrations of PB: Cu and PB: CuFe NCs were injected, followed by a control injection of deionized water. When exposed to an 808 nm laser at the power of 1.0 W cm^−2^, the temperature of aqueous solutions containing PB: CuFe NCs can reach 44.8, 50.2, 59.0, and 61.7, corresponding to the concentrations of 25 50, 100, and 200 μg mL^−1^, respectively. Under identical circumstances and concentration gradient, the temperature of PB: Cu NCs only increased from 23.7 °C to 26.5 °C with the increase in concentration from 25 to 200 μg mL^−1^. In comparison, the temperature of deionized water was just 23.1 °C (Figure 4B,E). The photothermal stability of PB: Cu and PB: CuFe NCs was assayed using an 808 nm laser with a power density of 1.0 W cm^−2^ for 10 min. There was little change in the photothermal cycle stability of PB: Cu NCs in this condition. For PB: CuFe NCs, however, the temperature remained a relative constant after five repeated irradiation cycles, increasing to 60.0 °C after each cycle (Figure 4C,F). 

Photothermal images of the PB: Cu and PB: CuFe NCs were captured using a thermal infrared camera. The photothermal images are shown in Figure 5A,B. It can be demonstrated that the temperature increment is related to the irradiation time and the concentration of the two samples. PB: Cu and PB: CuFe NCs could reach maximum temperatures of 26.5 °C and 61.7 °C, respectively, after being exposed to an 808 nm laser (1.0 W cm^−2^) for 10 min (Figure 5C). PB: CuFe NCs had a greater heating temperature than PB: Cu NCs at the same mass concentration. The increase in the absorbance at 808 nm with the increasing concentrations of PB: Cu and PB: CuFe NCs indicates that the maximum temperature of the aqueous solution increases with the laser irradiation time continuously (Figure 5D). At 808 nm, the absorption value of PB: CuFe NCs is noticeably more significant than that of PB: Cu NCs. In order to analyze photothermal performance thoroughly, the photothermal conversion efficiency was calculated using Roper’s approach [36]. The photothermal conversion efficiency (*η*), calculated from the maximum steady-state temperature and the time constant (*τ*), presents a quantitative comparison of the light-to-heat conversion ability of PB: Cu and PB: CuFe NCs. Figure 5E,F display the cooling curves of PB: Cu and PB: CuFe NCs, and the photothermal conversion efficiency of PB: Cu and PB: CuFe NCs is 1.9% and 37.92%, respectively, calculated from this curve. Therefore, when exposed to 808 nm NIR light, PB: CuFe NCs exhibit much better NIR light absorption and heat energy production.

Usually, a change in the parameters of the atoms may affect the parameters of the cell due to the change in the corresponding band structure [37]. PB: Cu and PB: CuFe NCs exhibit variable optical properties via the modulation of carrier density depending on PB’s internal dominant atom size [38]. The concentration of [Fe(CN)_6_]^4−^ vacancy decreases when more Cu^2+^ occupies the lattice position to form Fe-C = N–Zn [39]. Accordingly, it can be concluded that the existence of Cu^2+^ in the PB lattice alters the cyanide bond’s orbital energy and electron density. Additionally, the photothermal characteristics are not driven by the bandgap-narrowing effect but by the significant absorption difference between PB: Cu and PB: CuFe NCs at 808 nm. Consequently, PB: Cu NCs have a low photothermal conversion efficiency, while PB: CuFe NCs have a high efficiency.

### 3.3. Catalytic Properties of PB:Cu and PB:Fe NCs

A typical colorimetric experiment using o-phenylenediamine (OPDA) was conducted to assess the catalytic activity of PB: CuFe NCs. In the presence of H_2_O_2_, PB: CuFe NCs may catalyze the conversion of OPDA into a yellow solution of oxOPDA (Figure 6G). Additionally, adding PB: CuFe NCs to the H_2_O_2_ + OPD solution results in a yellow solution with a notable absorption peak at 417 nm (Figure 6A). While no noticeable color change was detected in the OPDA, OPDA + H_2_O_2_, and H_2_O_2_ + PB: CuFe NCs groups, at a given concentration of H_2_O_2_ and OPDA, the maximum absorbance of the reaction solution at 417 nm gradually increases with the increase in the concentration of PB: CuFe NCs. The experiment shows that a higher concentration of PB: CuFe NCs leads to more significant catalytic activity of the reaction solution and boosts OPDA oxidation (Figure 6C). The number of active sites increases with the amount of catalysts, allowing a much more extensive conversion of H_2_O_2_ to ·OH. The changes in absorption value in 10 min at the given concentration of PB: CuFe NCs were measured. With the increase in the reaction time, the absorption value at 417 nm approached saturation (Figure 6B). The steady-state catalytic kinetics of PB: CuFe NCs were further evaluated at room temperature. The variation in absorbance (417 nm) of the reaction solution at pH 5.5, 6.4, and 7.2 is shown in Figure 6D–F. There is a linear correlation between the absorbance and H_2_O_2_ concentration, and the absorbance reaches a higher value under micro-acidic conditions. Also, the maximum velocity (*V_max_*) and Michaelis–Menten constant (*K_m_*) were calculated using Equation (5). The *V_max_* and *K_m_* of PB: CuFe at pH 5.5, 6.4, and 7.2 was determined to be 0.914, 1.169, 1.324 mM s^−1^, and 1.021, 1.198, 1.411 mM, respectively (Figure 6H,I). PB: CuFe NCs have a higher affinity for H_2_O_2_ in the presence of *K_m_*, particularly at a pH of 5.5. PB: CuFe NCs exhibit much higher *V_max_* and *K_m_* values, indicating that they are more active as catalysts.

Instead of a heat source, 808 nm laser irradiation was used to determine whether heat aids the process of catalysis. Due to the photothermal effect, the absorbance of oxOPDA increases considerably after 5 min exposure to an 808 nm laser (Figure 6J). The absorbance of oxOPDA at pH 5.5 was further followed in comparison with PB: CuFe NCs, PB: CuFe NCs in a dark environment of 55 °C, and PB: CuFe NCs after 5 min of 808 nm laser irradiation (Figure 6K). The laser group and the heating group were significantly superior to the material group. These suggest that laser irradiation at 808 nm creates heat synergistically for PB: CuFe NCs, resulting in the enhancement of the catalytic capabilities. The reaction mechanism is as follows:

The PB:Fe NCs are catalyzed to generate the excited PB: CuFe NCs* [40,41]:(6)$$[K^{+}-T(\mathrm{CN}{)}_{6}{\mathrm{Fe}}^{2+}]\;\to \;[K^{+}-{\mathrm{Fe}}^{2+}(\mathrm{CN}{)}_{6}T]*$$

Metastable [K^+^ − Fe^2+^(CN)_6_T]* may oxidize H_2_O_2_ and liberate oxygen:(7)$$[K^{+}-\;{\mathrm{Fe}}^{2+}(\mathrm{CN}{)}_{6}T]*\;+\;H_{2}O_{2}+K^{+}\;\to \;[K^{+}-\;{\mathrm{Fe}}^{2+}(\mathrm{CN}{)}_{6}T*]\;+\;O_{2}\;+\;2H^{+}$$

The product [K^+^ − Fe^2+^(CN)_6_T*] in the reducing state can convert H_2_O_2_ to OH:(8)$$[K^{+}-{\mathrm{Fe}}^{2+}(\mathrm{CN}{)}_{6}T*]+H_{2}O_{2}\;\to \;-[K^{+}-\;{\mathrm{Fe}}^{2+}(\mathrm{CN}{)}_{6}T]\;+\;K^{+}\;+\;\mathrm{OH}\;+\;{\mathrm{OH}}^{-}$$
where T stands for the reaction ions. The conversion of Fe^3+^ and Fe^2+^ grants PB: CuFe NCs excellent catalytic properties. Further, the photothermal reaction accelerates the charge transfer between Fe^3+^ and Fe^2+^, accelerating the generation of ·OH. Hence, Cu: Fe NCs exhibit outstanding catalytic performance with the aid of the photothermal effect, indicating the synergistic impact of photothermal and catalysis.

## 4. Conclusions

Copper-based PB nanocubes and ion-doped copper-based PB nanocubes in the presence of PVP and HCl were fabricated using a facile aqueous synthetic method. Through manipulating the reaction time, nanocubes were formed at the initial stage, and then a void-like structure for PB: Cu NCs and collapsed structure for PB: CuFe NCs was observed. By substituting Cu^2+^ with Fe^2+^, PB: Cu and PB: CuFe NCs could have tunable plasmonic characteristics with a broad-spectrum range from 400 to 700 nm. The photothermal performance demonstrated that PB: CuFe NCs have a higher photothermal conversion capability, and the photothermal conversion efficiency was 37.92%. Catalysis performance showed that PB: CuFe NCs have higher values of *V_max_* and *K_m_* under micro-acidic circumstances, due to metal ion redox conjugates. The enhancement of the catalytic efficiency of PB: CuFe NCs with the 808 nm laser further confirmed the synergistic interaction between photothermal and catalytic capabilities. It is expected that PB: CuFe NCs could offer significant potential in current photothermal agents and catalysts.

## Figures and Tables

**Figure 1 nanomaterials-13-01897-f001:**
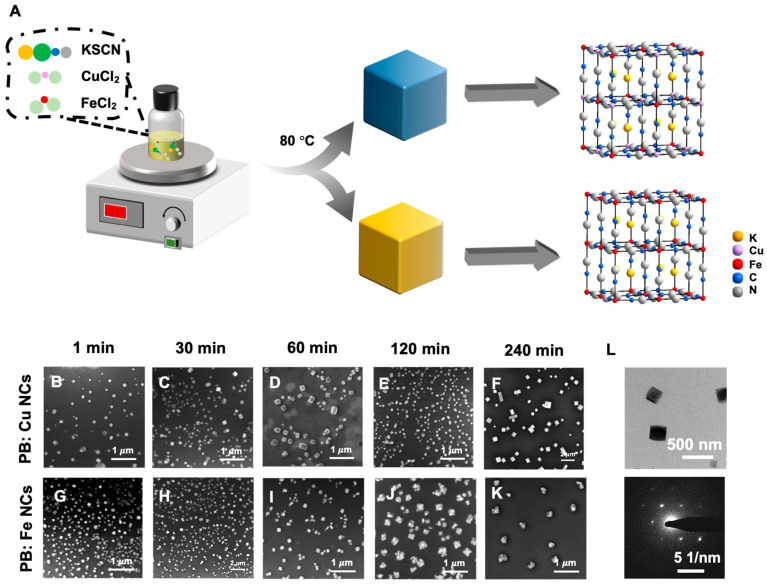
(**A**) Schematic illustration the synthesis process of PB: Cu and PB: CuFe NCs and SEM images of PB: Cu and PB: CuFe NCs synthesized with 0.15 mM CuCl_2_ and 0.15 mM FeCl_2_ during the different reaction times. (1 min: (**B**,**G**); 30 min: (**C**,**H**); 60 min: (**D**,**I**); 120 min: (**E**,**J**); 240 min: (**F**,**K**)). (**L**) TEM image and selected area electron diffraction (SAED) patterns of PB: CuFe NCs during the reaction time at 1 min.

**Figure 2 nanomaterials-13-01897-f002:**
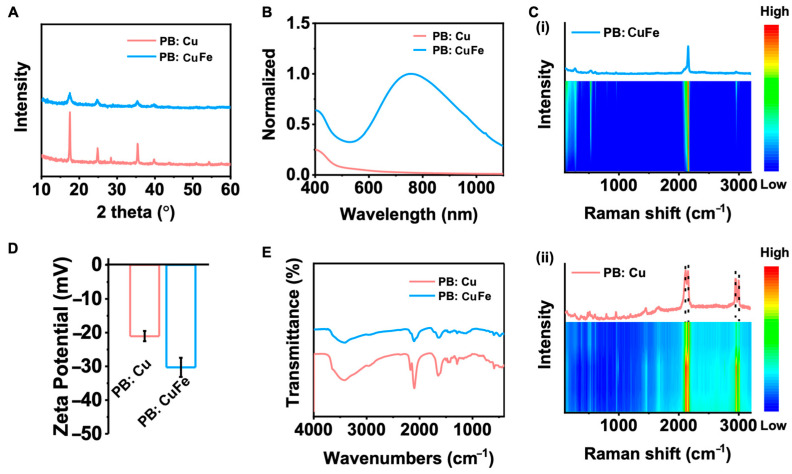
Structure characterization of PB: Cu and PB: CuFe NCs. (**A**) XRD patterns of the prepared PB: Cu and PB: CuFe NCs. (**B**) UV−vis spectra of PB: Cu and PB: CuFe NCs. (**C**) Time-dependent Raman spectra over a 60 min continuous exposure to a 532 nm laser (**i**) PB: CuFe NCs; (**ii**) PB: Cu NCs. (**D**) Zeta potentials of PB: Cu and PB: CuFe NCs. (**E**) FTIR spectra show that the NPs’ functional groups were successfully modified by measuring their distinctive peaks.

**Figure 3 nanomaterials-13-01897-f003:**
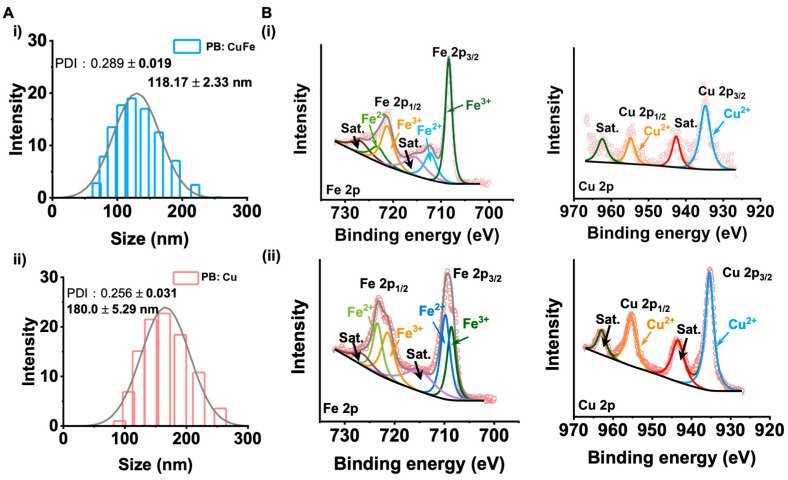
(**A**) The hydrodynamic size of (**i**) PB: Cu and (**ii**) PB: CuFe NCs. (**B**) The XPS of PB: CuFe NCs (**i**) and PB: Cu NCs (**ii**): Fe 2p, Cu 2p.

**Figure 4 nanomaterials-13-01897-f004:**
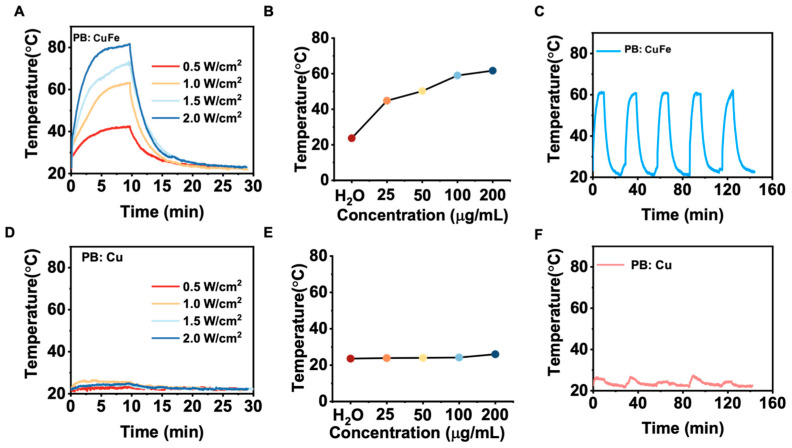
The photothermal properties of the PB: Cu and PB: CuFe NCs after 808 nm laser irradiation for 10 min at the power intensity of 1 W cm^−2^. The photothermal heating curve of the PB: CuFe (**A**) and PB: Cu (**D**) NCs with different power densities: 0.5, 1.0, 1.5, and 2.0 W cm^−2^. Concentration-dependent temperature changes in the PB: CuFe (**B**) and PB: Cu NCs (25, 50, 100, and 200 μg mL^−1^) (**E**). Photothermal stability of the (**F**) PB: Cu and (**C**) PB: CuFe NCs over five cycles of heating/cooling processes in which the heating was performed by irradiation with an 808 nm NIR laser (1 W cm^−2^) for 5 min.

**Figure 5 nanomaterials-13-01897-f005:**
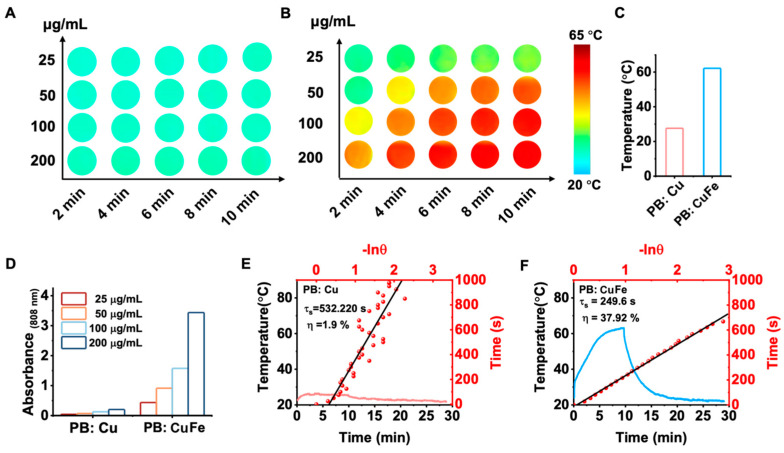
Photothermal images of the PB: Cu (**A**) and PB: CuFe (**B**) NCs after 808 nm laser irradiation for 10 min at the power intensity of 1 W cm^−2^ were recorded by a thermal infrared camera every 2 min. (**C**) The maximum steady-state temperature of the PB: Cu and PB: CuFe NCs. (**D**) Concentration-dependent absorbance changes in the PB: Cu and PB: CuFe NCs. The photothermal heating curve of the PB: Cu (**E**) and PB: CuFe (**F**) NCs cooling curve after turning off the laser.

**Figure 6 nanomaterials-13-01897-f006:**
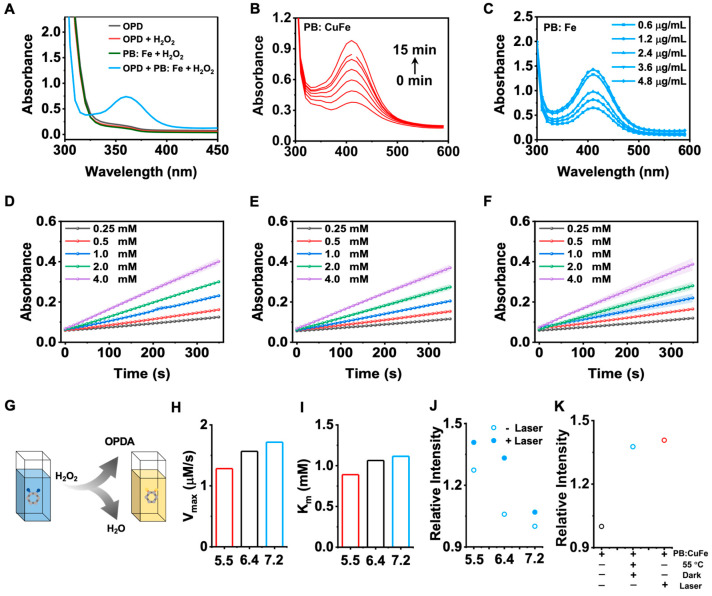
Characterizations of the OPDA catalytic activity of the PB: CuFe NCs. (**A**) UV–vis absorption spectra of the OPDA, OPDA + H_2_O_2_, H_2_O_2_ + PB: CuFe NC, and OPDA + H_2_O_2_ + PB: CuFe NC groups. (**B**) UV–vis absorption spectra of the PB: CuFe NCs during the 15 min. (**C**) UV–vis absorption spectra of the PB: CuFe NCs with the addition of various concentrations. Time-dependent absorbance changes at 417 nm as a result of the catalyzed oxidation of OPDA with the PB: CuFe NCs and H_2_O_2_ concentrations at pH 5.5 (**D**), 6.4 (**E**), and 7.2 (**F**). (**G**) Schematic illustration of the mechanism of the PB: CuFe NCs. The maximum reaction rate (*V_max_*) (**H**) and Michaelis–Menton constant (*K_m_*) (**I**) of the PB: CuFe NCs. (**J**) Influence of photothermal properties on catalytic performance with or without 808 nm laser irradiation for 5 min at the power intensity of 1 W cm^−2^. (**K**) Influence of photothermal properties with different groups at pH 5.5: PB: CuFe NCs, PB:CuFe NCs + 55 °C+ dark; PB:CuFe NCs + laser.

## Data Availability

The data that support the findings of this study are available from the corresponding authors upon reasonable request.

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
