# Peer review of "Photothermal and Catalytic Performance of Multifunctional Cu-Fe Bimetallic Prussian Blue Nanocubes with the Assistance of Near-Infrared Radiation"

_nanomaterials, 2023, doi:10.3390/nano13131897_

Round 1

Reviewer 1 Report

The authors have conducted a systematic study on the synthesis, characterization, and performance evaluation of Cu-Fe bimetallic Prussian Blue nanocubes. The manuscript is well-organized, presenting abundant data, and demonstrates superior performance compared to PB-Cu. The writing is of good quality, and the figures are excellently displayed. Based on these merits, I recommend this manuscript for publication in "nanomaterials." However, I have two concerns that should be addressed before publication:

1. In Figure 6, the lines or traces for B and C are unclear. It is difficult to differentiate between them as they share the same color and nearly identical dot shapes. Enhancing the clarity of the figures is essential to ensure proper interpretation. Please revise the figures accordingly.

2.   It would be valuable to obtain TEM data for the nanotubes, specifically to capture the crystal lattice for further analysis. This additional information could provide valuable insights into the structural properties of the nanotubes. I encourage the authors to include TEM data if possible.

Author Response

Response to Reviewer 1 Comments

We would first like to thank the reviewers for the constructive comments, which are very helpful for us to improve the manuscript. We have revised the manuscript by following the advices of the reviewers or by addressing the questions raised by the reviewers. The point-to-point response to all the questions raised is listed as follows. For your convenience, the page and line numbers of the locations of the revisions are included in brackets.

Point 1: In Figure 6, the lines or traces for B and C are unclear. It is difficult to differentiate between them as they share the same color and nearly identical dot shapes. Enhancing the clarity of the figures is essential to ensure proper interpretation. Please revise the figures accordingly.

Response 1: We have made the change according to the comment. The color of picture 6B has been changed to red, which is obviously different from picture 6C. (Page 10, Figure 6)

Point 2: It would be valuable to obtain TEM data for the nanotubes, specifically to capture the crystal lattice for further analysis. This additional information could provide valuable insights into the structural properties of the nanotubes. I encourage the authors to include TEM data if possible.

Response 2: We sincerely thank you for this suggestion. We have added TEM data and analysis in section 3.1: “Transmission electron microscopy (TEM) of PB: CuFe NCs revealed a solid cubic …”. (Page 4, Line 149-152; Page 5, Figure 1)

Reviewer 2 Report

This paper deals with the synthesis of Cu and Cu/Fe Prussian blue MOFs and study of their photothermal performance and catalytic oxidation of o-phenylenediamine (OPDA) with H2O2. In my opinion, the paper requires a major revision before the acceptance. My issues are the following:

11.  The Cu/Fe ratio in Cu/Fe PB is missing in the paper. The data about chemical composition of prepared compounds must be presented in the revised manuscript.

22. The optical absorption band at ca. 700 nm of Cu/Fe PB was wrongly attributed to the localized surface plasmon resonance (LSPR). Actually, LSPR is based on the collective excitation of electrons in metallic nanoparticles. This is not the case of Fe(II)-Fe(III) charge transfer band of Cu/Fe PB. This error must be corrected in the revised paper.

33.  The Ref. 40 is irrelevant because in this work there is no data about LSPR of PB particles.

44. According to the paragraph 2.3 the photothermal performance was calculated by the equation 1. However, in the Results and Discussions the data about the photothermal conversion efficiency are missing. The authors should provide these data calculated according to the equation 1 or remove the equation 1 from the text.

55. Regarding catalytic properties of Cu/Fe PB it is not clear whether the catalytic oxidation of OPDA is triggered by light excitation or by thermal excitation. The authors should provide the experiments under the heating at dark conditions and compare the results with the experiments under laser irradiation at the same temperatures.

Author Response

Response to Reviewer 2 Comments

We would first like to thank the reviewers for the constructive comments, which are very helpful for us to improve the manuscript. We have revised the manuscript by following the advices of the reviewers or by addressing the questions raised by the reviewers. The point-to-point response to all the questions raised is listed as follows. For your convenience, the page and line numbers of the locations of the revisions are included in brackets.

Point 1: The Cu/Fe ratio in Cu/Fe PB is missing in the paper. The data about chemical composition of prepared compounds must be presented in the revised manuscript.

Response 1: We have made corresponding supplement in view of this question in the article. (Page 2, Line 92-94)

Point 2: The optical absorption band at ca. 700 nm of Cu/Fe PB was wrongly attributed to the localized surface plasmon resonance (LSPR). Actually, LSPR is based on the collective excitation of electrons in metallic nanoparticles. This is not the case of Fe(II)-Fe(III) charge transfer band of Cu/Fe PB. This error must be corrected in the revised paper.

Response 2: We sincerely appreciate you for pointing out this mistake. We have revised the errors in the modified manuscript. (Page 5, Line 183-184, 190-192; Page 8, 281-283, 286-287; Page 9, 288; Page 13 line 454-455)

Point 3: The Ref. 40 is irrelevant because in this work there is no data about LSPR of PB particles.

Response 3: We sincerely appreciate you for pointing out this mistake. The wrong reference 40 was corrected. (Page 13, Line 464-466)

Point 4: According to the paragraph 2.3 the photothermal performance was calculated by the equation 1. However, in the Results and Discussions the data about the photothermal conversion efficiency are missing. The authors should provide these data calculated according to the equation 1 or remove the equation 1 from the text.

Response 4: Thank you for the constructive suggestion. The photothermal conversion efficiency was added: “Figures 5E and F display the cooling curves of PB: Cu and PB: CuFe NCs, and the photothermal conversion efficiency of PB: Cu and PB: CuFe NCs...”. Also, We annotated the calculated photothermal conversion efficiency in the corresponding Figure 5. (Page 8, Line 275-277; Page 9, Figure 5)

Point 5: Regarding catalytic properties of Cu/Fe PB it is not clear whether the catalytic oxidation of OPDA is triggered by light excitation or by thermal excitation. The authors should provide the experiments under the heating at dark conditions and compare the results with the experiments under laser irradiation at the same temperatures.

Response 5: We sincerely thank you for the constructive suggestion. We have made corresponding supplement in view of this question in the article. (Page 10, Line 323-327; Page 10, Figure 6; Page 11, Line 348-350)

Round 2

Reviewer 2 Report

The revised paper has been sufficiently improved and can be accepted for publication.